# On the Inner Horizon Instability of Non-Singular Black Holes

**Francesco Di Filippo** [1,*] , **Raúl Carballo-Rubio** [2] , **Stefano Liberati** [3,4,5] , **Costantino Pacilio** [6] **and Matt Visser** [7]

1   Center for Gravitational Physics, Yukawa Institute for Theoretical Physics, Kyoto University,
    Kyoto 606-8502, Japan
2   Florida Space Institute, University of Central Florida, 12354 Research Parkway, Partnership 1,
    Orlando, FL 32826, USA; raul.carballorubio@ucf.edu
3   SISSA—International School for Advanced Studies, Via Bonomea 265, 34136 Trieste, Italy; liberati@sissa.it
4   IFPU—Institute for Fundamental Physics of the Universe, Via Beirut 2, 34014 Trieste, Italy
5   INFN Sezione di Trieste, Via Valerio 2, 34127 Trieste, Italy
6   Dipartimento di Fisica, "Sapienza" Università di Roma & Sezione INFN Roma1, Piazzale Aldo Moro 5,
    00185 Roma, Italy; costantino.pacilio@uniroma1.it
7   School of Mathematics and Statistics, Victoria University of Wellington, P.O. Box 600,
    Wellington 6140, New Zealand; matt.visser@msor.vuw.ac.nz
*   Correspondence: francesco.difilippo@yukawa.kyoto-u.ac.jp

**Abstract:** Regular black holes represent a conservative model in which the classical singularity is replaced by a non-singular core without necessarily modifying the spacetime outside the trapping horizon. Given the possible lack of phenomenological signatures, it is crucial to study the consistency of the model. In this short work, we review the physical mechanism leading to the instability of the central core, arguing that that non-perturbative backreaction is non-negligible and must be taken into account to provide a meaningful description of physical black holes.

**Keywords:** regular black holes; singularity regularization; mass inflation instability

## 1. Introduction

Gravitational and electromagnetic observations provide remarkable experimental support for the existence of black holes, as described by the theory of general relativity (see [1] and references therein). On the other hand, standard results show that within general relativity, gravitational collapse unavoidably produces a singularity once a trapping region is formed [2]. The formation of singularities is theoretically unpleasant, as it signals the breakdown of validity of the theory. However, it is reasonable to expect that the singularity is regularized by quantum gravity effects, once they are consistently accounted for (see, however, ref. [3] for a discussion concerning different points of view on this issue).

In the following, restricting for simplicity to spherically symmetric spacetimes, we will study a conservative class of models describing non-singular black holes in which the singularity is replaced by a regular core. A static and spherically symmetric regular black hole metric can be parametrized as [4,5]

$$ds^2 = -e^{-2\phi(r)}F(r)dt^2 + \frac{dr^2}{F(r)} + r^2 d\Omega, \tag{1}$$

where $\phi(r)$ and $F(r)$ are two real functions. It will be sometimes convenient to introduce the notation

$$F(r) = 1 - \frac{2M(r)}{r} \tag{2}$$

where $M(r)$ is the Misner–Sharp mass [6]. The absence of curvature singularities implies that $M(r)$ vanishes at least as $r^3$ in the limit $r \to 0$. On the other hand, asymptotic flatness implies that $F(r) \to 1$ in the asymptotic region $r \to \infty$. Therefore, the function $F(r)$ must have an even number of zeros. For simplicity, but without loss of generality, we will

consider the case in which $F(r)$ has only two zeros located at $r = r_\pm$, in correspondence of the inner and outer horizon.

It is straightforward to check the surface gravity at the horizons is given by

$$\kappa_\pm = \left. \frac{e^{-\phi(r)}}{2r} \frac{d}{dr} F(r) \right|_{r=r_\pm}.$$
(3)

In particular, $\kappa_- < 0$ and $\kappa_+ > 0$. This implies that, while at the outer horizon we have an exponential peeling of the outgoing null rays, at the inner horizon we have an exponential focusing of the outgoing null rays. This behavior is the root of the instability of the inner horizons within general relativity. In this work, we are going to study the instability of inner horizons beyond general relativity in order to clarify which aspects are general consequences of a purely geometrical treatment and which aspects would require the dynamical field equations of a specific theory.

This work summarizes the analysis of [7,8] and is organized as follows. Section 2, mainly based on ref. [7], studies the mass inflation instability arising when the spacetime is perturbed by two null shells that cross close to the inner horizon. Following ref. [8], in Section 3 the physical perturbation is slightly modified as one of the shells is substituted by a continuous flux of energy. Finally, Section 4 contains the main conclusions and discusses the most commonly asked questions.

## 2. Double Null Shell

Let us now consider a perturbation of the background spacetime in order to study the stability of the inner horizon.

The specific type of perturbation we are going to consider is the same that was studied in [9]. As depicted in Figure 1, we consider two null shells, $\Sigma_3$ and $\Sigma_4$, that intersect each other in a two surface $S$ (that we will eventually move close to the inner horizon) of radius $r_0$ and produce two other null shells, $\Sigma_1$ and $\Sigma_2$. The shells divide the spacetime into four regions. Let us denote the four regions $A$, $B$, $C$, and $D$, and the vectors tangent to the shells $l_{(i)}$. The mass inflation instability develops in the region A between the two null shells $\Sigma_1$ and $\Sigma_2$. The result follows straightforwardly from [9] (see also [10] for a more pedagogical description). However, ref. [9] focuses on general relativity, therefore it is useful to repeat the analysis here to explicitly show the underlying assumptions. We will see that the result can be obtained via purely geometrical considerations without specifying the dynamical equations of the theory.

The spacetime under consideration contains curvature singularities due to the presence of the thin shells [11], which are objects of zero width but finite energy. These singularities have a clear physical interpretation and they would go away in a less idealized situation. On the other hand, we need to assume regularity conditions to ensure the absence of singularities which do not have any physical interpretation. In particular, Israel's first junction condition [11], on purely geometrical grounds, tells us that there must be a well defined notion of induced metric $\sigma_{ab}^i$ on each shell $\Sigma_i$. This means that projecting either one of the four dimensional metrics on the two sides of each shell results in the same two dimensional metric.

We also need to assume that the spacetime is well behaved on $S$ (for instance, we do not allow for the presence of conical singularities). As a consequence, each point on $S$ can be covered by a coordinate chart in which the metric is continuous and (piecewise) differentiable. Using these coordinates, together with the fact that a two-surface embedded in a four-dimensional spacetime has only two orthogonal null directions, we find that at $S$, $l_{(3)}$ is parallel to $l_{(2)}$ and $l_{(4)}$ is parallel to $l_{(1)}$

$$l_{(3)}^\mu = \alpha l_{(2)}^\mu, \qquad l_{(4)}^\mu = \beta l_{(1)}^\mu.$$
(4)

These relations hold only in the particular coordinate chart in which the metric is continuous and (piecewise) differentiable along $S$. However, it is trivial to use them to obtain the coordinate invariant relations

$$\left(l_{(1)} \cdot l_{(2)}\right)\left(l_{(3)} \cdot l_{(4)}\right) = \left(l_{(1)} \cdot l_{(3)}\right)\left(l_{(2)} \cdot l_{(4)}\right). \tag{5}$$

These relations constitute the first building block of the analysis. An additional relation is obtained by considering the trace of extrinsic curvature

$$K^{(i)} = \sigma^{ab}_{(i)} K^{(i)}_{ab} = \sigma^{ab}_{(i)} \mathcal{L}_{l_{(i)}} \sigma^{(i)}_{ab} = \sigma^{ab}_{(i)} l^\alpha \partial_\alpha \sigma^{(i)}_{ab} = \frac{2}{r} l^\alpha_{(i)} \partial_\alpha r, \tag{6}$$

where $\mathcal{L}$ denotes the Lie derivative, and capital Latin indexes run from 1 to 2, while Greek indexes run from 0 to 3. The last step follows directly from the fact that we are considering spherical shells.

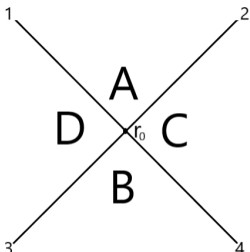

**Figure 1.** Two null shells cross in a two surface of radius $r_0$ dividing the spacetime into four regions. We are particularly interested in the situation in which $r_0$ is very close to the inner horizon.

Due to the null nature of the shells, the orthogonal vector is also tangential to the shell, and the extrinsic curvature is given by the tangential derivative of the metric along the shells. Therefore, the extrinsic curvature has to be continuous across the shell, and it cannot depend on which region of spacetime is used in the computation [11,12]. This would not be true if the shells were timelike, as the extrinsic curvature would not be well defined, as it would depend on the four dimensional metric which is different in the two side of the shell. Now, we can rewrite Equation (5)

$$\frac{K^{(1)} K^{(2)} K^{(3)} K^{(4)}}{\left(l_{(1)} \cdot l_{(2)}\right)\left(l_{(3)} \cdot l_{(4)}\right)} = \frac{K^{(1)} K^{(2)} K^{(3)} K^{(4)}}{\left(l_{(1)} \cdot l_{(3)}\right)\left(l_{(2)} \cdot l_{(4)}\right)} \tag{7}$$

or, grouping the terms in a different way

$$\left(\frac{K^{(1)} K^{(2)}}{l_{(1)} \cdot l_{(2)}}\right)_A \left(\frac{K^{(3)} K^{(4)}}{l_{(3)} \cdot l_{(4)}}\right)_B = \left(\frac{K^{(1)} K^{(3)}}{l_{(1)} \cdot l_{(3)}}\right)_D \left(\frac{K^{(2)} K^{(4)}}{l_{(2)} \cdot l_{(4)}}\right)_C \tag{8}$$

where the index refers to the region of spacetime we need to use in order to evaluate the quantities in the bracket. Here, it is clear why it is important that the extrinsic curvature is continuous across the shell. Substituting the explicit value of the extrinsic curvature, we have

$$\left(\frac{K^{(1)} K^{(2)}}{l_{(1)} \cdot l_{(2)}}\right)_A = \frac{4}{r^2} \frac{l^\alpha_{(1)} \partial_\alpha r l^\beta_{(2)} \partial_\beta r}{l_{(1)} \cdot l_{(2)}} = \frac{4}{r^2} \frac{l^\alpha_{(1)} l^\beta_{(2)} \delta^r_\alpha \delta^r_\beta}{l_{(1)} \cdot l_{(2)}} \tag{9}$$

Making use of the completeness relations

$$g^A_{\alpha\beta} = \sigma^{(1)}_{\alpha\beta} + \frac{l_{(1)\alpha} l_{(2)\beta}}{l_{(1)} \cdot l_{(2)}} + \frac{l_{(2)\alpha} l_{(1)\beta}}{l_{(1)} \cdot l_{(2)}}, \qquad g^B_{\alpha\beta} = \sigma^{(4)}_{\alpha\beta} + \frac{l_{(3)\alpha} l_{(4)\beta}}{l_{(3)} \cdot l_{(4)}} + \frac{l_{(4)\alpha} l_{(3)\beta}}{l_{(3)} \cdot l_{(4)}},$$

$$\tag{10}$$

$$g^C_{\alpha\beta} = \sigma^{(2)}_{\alpha\beta} + \frac{l_{(2)\alpha} l_{(4)\beta}}{l_{(2)} \cdot l_{(4)}} + \frac{l_{(4)\alpha} l_{(2)\beta}}{l_{(2)} \cdot l_{(4)}}, \qquad g^D_{\alpha\beta} = \sigma^{(3)}_{\alpha\beta} + \frac{l_{(1)\alpha} l_{(3)\beta}}{l_{(1)} \cdot l_{(3)}} + \frac{l_{(3)\alpha} l_{(1)\beta}}{l_{(1)} \cdot l_{(3)}},$$

we obtain

$$\left(\frac{K^{(1)}K^{(2)}}{l_{(1)}\cdot l_{(2)}}\right)_A = \frac{2}{r^2}\left(g^{A\ \alpha\beta} - \sigma^{(1)\ \alpha\beta}\right)\delta^r_\alpha\delta^r_\beta = \frac{2}{r^2}g^{A\ rr}. \tag{11}$$

Considering that the coordinate charts in which the metrics in the four regions are presented all take the form (1) (possibly with different functions $F(r)$ and $\phi(r)$), we find

$$F_A(r_0)F_B(r_0) = F_C(r_0)F_D(r_0) \tag{12}$$

which goes by the name of DTR relation [13,14]. We can rewrite it in terms of the Misner–Sharp quasilocal mass obtaining

$$M_A = M_B + M_{\text{in}}(r_0) + M_{\text{out}}(r_0) - 2\frac{M_{\text{in}}(r_0)M_{\text{out}}(r_0)}{r_0 F_B(r_0)}. \tag{13}$$

where $M_{\text{in}}(r_0) = M_C(r_0) - M_B(r_0)$ and $M_{\text{out}}(r_0) = M_D(r_0) - M_B(r_0)$ can be interpreted as the mass of the ingoing and outgoing null shells. Very close to the inner horizon, $F_B$ goes to zero due to the definition of the inner horizon, and $M_{\text{in}}$ goes to zero as well because we are probing the late time perturbations. For the time being, we will assume that the perturbations follow the Price's law [15]

$$M_{\text{in}} \propto v^{-\gamma} \tag{14}$$

with $\gamma$ a positive constant. On the other hand, the behavior of $F_B$ in the vicinity of the inner horizon can be derived from the metric (see [7] for details), obtaining

$$F_B \propto e^{-|\kappa_-|v}. \tag{15}$$

Inserting Equations (14) and (15) into Equation (13), we obtain

$$M_A \propto v^{-\gamma}e^{|\kappa_-|v}. \tag{16}$$

Therefore, close to the inner horizon, a small perturbation has a huge backreaction on the geometry, signalling the presence of an instability.

### 3. Modified Ori Problem

Let us now consider a slightly different type of perturbation in which we still have an outgoing null shell, but the ingoing null shell is substituted by a continuous stream of energy. Such configuration was initially considered by Ori [16] to study the instability of Reissner–Nordström black holes. As indicated in Figure 2, the spacetime is now divided in two regions, $\mathcal{R}^-$ and $\mathcal{R}^+$, by the ingoing null shell $\Sigma$.

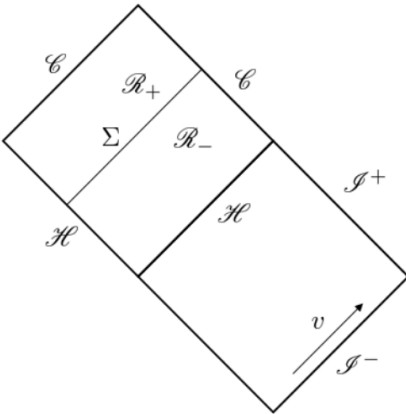

**Figure 2.** Relevant quadrant of the Penrose diagram of a regular black hole with two horizons.

We assume that, in the two regions, the metric takes the same functional form of a regular black hole solution in which the Misner–Sharp mass is now time dependent.

$$ds^2 = -f_\pm(v_\pm, r)dv_\pm^2 + 2dv_\pm dr + r^2 d\Omega^2. \tag{17}$$

with

$$f(v_\pm, r) = 1 - \frac{2M_\pm(v_\pm, r)}{r}. \tag{18}$$

Furthermore, we consider that the $v$ dependence enters via the variable $m(v)$ that coincides with the asymptotic value of the Misner–Sharp mass (Note that the + region does not extend up to $r \to \infty$ and the corresponding limit is thus intended as a formal mathematical definition of the function $m_+$).

$$M_\pm(v_\pm, r) = M_\pm(m_\pm(v_\pm), r), \qquad m_\pm(v_\pm) = \lim_{r \to \infty} M_\pm(v_\pm, r). \tag{19}$$

The value of $m_-(v_-)$ is fixed once again by the Price law

$$m_- = m_0 - \beta v^{-\gamma}, \tag{20}$$

with $\beta$ and $\gamma$ positive constants, and we have renamed $v_- \equiv v$.

We now need to determine the behavior of $M_+$ close to the inner horizon. To this end, we consider the junction condition at the shell [12]

$$\left[ T_\mu{}^\nu s^\mu s_\nu \right] = 0 \tag{21}$$

where $T_\mu{}^\nu$ is the *effective* stress energy tensor that is obtained from the Einstein equations, $s^\mu = (2/f_\pm, 1, 0, 0)$ is the outgoing null vector normal to the shell, and the square brackets indicate the discontinuity across the shell. Straightforward manipulations [8] result in

$$\frac{1}{f_+} \frac{\partial M_+}{\partial v}\bigg|_{r=R(v)} = \frac{1}{f_-} \frac{\partial M_-}{\partial v}\bigg|_{r=R(v)}, \tag{22}$$

in which $R(v)$ denotes the radial location of the shell.

Contrary to the analysis in the previous section, it is no longer possible to solve this equation for a generic expression of the Misner–Sharp mass, and we need to specify the relation between $M_\pm(v, r)$ and its asymptotic value $m_\pm(v)$. The details of the analysis are explained in ref. [8]; here, we simply state the main results. If the Misner–Sharp mass depends on the asymptotic mass linearly, i.e.,

$$M(v, r) = g_1(r)m(v) + g_2(r), \tag{23}$$

the late time behaviour of the solution of Equation (22) develops an exponential growth

$$M_+ \sim \frac{e^{|\kappa_-|v}}{v^{\gamma+1}}. \tag{24}$$

Geometries in this class include, for instance, Reissner–Nordström black hole and Bardeen's regular black hole [17]. On the other hand, for more generic mass functions, the late time behavior is not necessarily exponential. For instance, for Hayward's regular black hole [4] we find

$$M_+ \propto |\kappa_-| \frac{v^{\gamma+1}}{\beta}. \tag{25}$$

It is curious to note that the instability is slower for larger perturbations. This counter-intuitive result and the difference with the late time behavior of the Misner–Sharp mass obtained in the previous section is rooted in the fact that the ingoing flux modifies the

location of the inner horizon. If the absorption rate is high enough and constant this effect could, in principle, partially tame the mass inflation instability.

However, let us stress that the polynomial instability is always preceded by an exponential phase. At the end exponential instability phase the backreaction on the geometry is very big and the linear approximation used in this approach cannot be trusted anymore. For instance, in the case of Hayward's metric, the transition between the exponential and polynomial phase occurs when the ratio between the Misner–Sharp mass in the interior region $M_+$ and the the initial mass $m_0$ is given by

$$\frac{M_+}{m_0} \sim m \frac{v_0^{\gamma+1}}{6\beta} \frac{m_0}{\ell} \gg 1 \,. \tag{26}$$

Therefore, the polynomial behavior cannot be trusted as it is predicted by the model only after the end of the regime of validity of the model itself. As a consequence, in the relevant regime of validity, the behavior of the instability is still described by an exponential growth.

## 4. Discussion and Answer to the Most Common Questions

In this work, we have discussed the interplay between the inner horizon instability and the viability of regular black holes as a resolution to the singularity problem. As these notes are based on a series of talks that the authors have given several times, we believe it can be beneficial for the reader to conclude by addressing some of the most commonly asked questions and the main points that can be possibly misleading.

### 4.1. What Are the Main Differences between the Two Approaches Described in the Two Sections? Why Are the Results Different?

We have described two different approaches to obtaining results in qualitative agreement with each other. The main difference consists in the use of two different types of perturbations. In the first approach, we have considered an ingoing and an outgoing null shell crossing close to the inner horizon, while in the second approach the ingoing null shell is replaced with a continuous flux of matter. They represent two simplified models describing physically distinct perturbation configurations: the two-shell approach is well suited to describe a configuration in which the black holes absorb in a discontinuous way, whereas, if the black hole accretes at constant rate, the continuous flux approach accounts for the continuous displacement of the location of the inner horizon. We stress that both perturbations configurations are realistic in the vicinity of the inner horizon, where the thin shell approximation is reasonable due to the intense blue shift.

### 4.2. Does the Polynomial Growth Imply That the Instability Is Slowed Down?

Equation (25) shows that, for a Hayward regular black hole, the late time growth of the perturbation is polynomial rather than exponential. We might be tempted to conclude that the instability is slowed down in this case. There are two reasons for which we believe that this conclusion is not correct. First of all, an astrophysical black hole spacetime should be stable under a generic perturbation, not only for a constant accretion rate. Therefore, the result obtained considering two null shells would be already enough to show that the instability is exponential. Furthermore, as explained in the text, even in the assumption of constant absorption rate, the polynomial phase is always preceded by an exponential phase. At the end of the exponential phase, the linear approximation used in this analysis is no longer valid. Therefore, we believe that we should not give physical relevance to the polynomial growth, as it is only predicted by the model after the end of validity of the model itself.

### 4.3. Does the Cosmological Constant Play Any Role?

To date, we have not considered the presence of the cosmological constant. There is a very intuitive reason behind this choice. The instability is generated close to the

inner horizon and the value of the cosmological constant does not affect the geometry in this region. On the other hand, a series of work on the validity of the strong cosmic censorship [18–21] seems to challenge the validity of this intuitive reasoning. These works show that the cosmological constant plays a crucial role in the study of the stability of the inner horizon of a Reissner–Nordström or Kerr black hole. In fact, in the presence of a cosmological constant, the asymptotic behavior of the geometry is modified, and the function *F* reads

$$F(r) = 1 - \frac{2M(r)}{r} + \Lambda r^2 \,. \tag{27}$$

As well as the inner and outer horizon, it is now also present on a cosmological horizon located at approximately $r_c \sim \Lambda^{-1/2}$. It can be shown [22] that the difference in the asymptotic behavior changes the late time behavior of the perturbations which, rather than the Price law (14), now follow an exponential fall off

$$m_{\mathrm{in}} \propto e^{-\omega_I v}, \tag{28}$$

where $\omega_I$ is the imaginary part of the least dumped quasinormal mode. For both the physical configurations in Sections 2 and 3, this fall off will tame the instability if $\omega_I > |\kappa_-|$. However, this does not imply that an arbitrary small cosmological constant can tame the inner horizon instability for two reasons. First of all, we expect the inner horizon to be located in a region where quantum gravity effects are dominant. Therefore, we expect that the surface gravity is of order of the Planckian curvature, leading $\omega_I < |\kappa_-|$. More importantly, the fall off of the perturbation will differ from the Price law only at very late time. From the physical point of view, it is easy to estimate this time. If the cosmological constant is very small, the perturbations need to reach regions of spacetime sufficiently close to the cosmological horizon in order to see any deviation from the Price law. Therefore, we can estimate the time up to which the fall-off of the perturbations fallow the Price law, even in the presence of a cosmological constant such as $v_c \sim r_c \sim \Lambda^{-1/2}$. For the cosmological constant of our universe, this timescale is so large that for any realistic initial conditions the instability has already developed. This shows why the cosmological constant has to be taken into consideration in the study of the strong cosmic censorship, but can be safely discarded in our analysis. In fact, for the strong cosmic censorship to hold in general relativity, the inner horizons of Kerr or Reissner-Nordström black holes have to be unstable for any initial conditions, even a strongly fine tuned one. On the other hand, for regular black holes to be a viable resolution of the singularity problem, the inner horizon has to be stable under any generic realistic perturbation.

### 4.4. Is It Reasonable to Consider Only the Late Time Behavior of the Perturbation?

In the text we have assumed the Price law, which only describes the late time fall off of the perturbation. This choice is justified by the fact that the inner horizon is located at infinity for the retarded time *v*. However, as pointed out in [23], astrophysical black holes continue to accreate matter for a very long time after their formation, and the instability might develop before reaching the inner horizon, thus before the perturbations start to follow the Price law. We certainly agree with this statement; however, the logic of our work is to show that regular black holes geometries cannot be trusted as the end point of singularity regularization. To this end, it is enough to show that an instability develops even in the idealized situation in which only the Price tail of the perturbation is present. Whether an instability develops even before the regime in which the Price tail is dominant is irrelevant for our conclusions.

### 4.5. Analogue Black Holes Seem to Have a Stable Inner Horizon, Why Does This Analysis Not Apply to Them?

Analogue black holes [24] allow us to mimic and study some properties of gravitational black holes in tabletop experiments. Analogue geometries have an inner horizon which appears to be stable over a timescale that allows for the detection of stimulated Hawking

radiation [25]. It is not clear if these geometries are truly stable or only metastable (in fact, in the setup of [25], the inner horizon moves towards the outer horizon, which might signal an instability). The analysis presented here cannot be immediately applied to analogue systems because these systems have modified dispersion relations, so there is no exponential blueshift close to the inner horizon and there is no reason to assume that null thin shells constitute a reasonable approximation to a real perturbation. A first attempt to study the stability of Cauchy horizons in the presence of modified dispersion relations was carried out in [26], where it was shown that warp drives instability is still present, but it can be slowed down by some specific choices of modified dispersion relation. However, to properly address the instability of regular black holes in Lorentz violating theories, we would also need to take into account the extra structure associated with these objects, e.g., studying the stability of the universal trapping horizon [27].

### 4.6. How Is It Possible to Prove the Presence of an Instability without Specifying the Dynamics of the Theory?

One of the main sources of confusion is related to the fact that the results obtained are apparently too strong given the little physical assumptions that have been made. In fact, it is important to stress that it is impossible to prove the formation of a physical singularity without specifying the field equation of the theory. Let us stress that this is *not* what we have proven. We have shown that an arbitrary small perturbation has an exponential effect on the geometry and leads to an arbitrary large growth of the Misner–Sharp mass. It is very interesting that this statement can be proved with purely kinematical arguments.

### 4.7. What Are the Main Conclusions of the Analysis?

This analysis shows that a wide class of regular black holes geometries is unstable once idealized but realistic perturbations are taken into account. This result is very general, as it does not rely on the dynamical equations of the theory. The fact that, when considering two different types of perturbation, we obtain qualitatively similar results indicates that the instability is a consequence of the geometrical settings rather than the specific type of perturbation.

In the context of general relativity, this instability usually leads to the formation of a physical singularity. However, without specifying the relevant dynamical equations, we cannot reach a definitive conclusion, and it is reasonable to expect that a full theory of quantum gravity will not produce a singularity. The end-point of the instability could be one of the other classes of non-singular geometries described in refs. [28,29] (see also [30]). Therefore, while the analysis summarized here poses serious questions regarding the viability of regular black holes as a resolution to the singularity problem, they do not imply that a quantum gravity model predicting regular black holes is not viable. In fact, our results should be taken as a strong motivation to study the nonlinear problem in specific quantum gravity frameworks.

**Author Contributions:** All authors contributed equally. All authors have read and agreed to the published version of the manuscript.

**Funding:** FDF acknowledges financial support by Japan Society for the Promotion of Science Grants-in-Aid for international research fellow No. 21P21318. SL acknowledges funding from the Italian Ministry of Education and Scientific Research (MIUR) under the grant PRIN MIUR 2017-MB8AEZ. CP acknowledges: financial support provided under the European Union's H2020 ERC, Starting Grant agreement no. DarkGRA–757480; support under the MIUR PRIN and FARE programmes (GW-NEXT, CUP: B84I20000100001); support from the Amaldi Research Center funded by the MIUR program "Dipartimento di Eccellenza" (CUP: B81I18001170001). MV was supported by the Marsden Fund, via a grant administered by the Royal Society of New Zealand.

**Institutional Review Board Statement:** Not applicable.

**Informed Consent Statement:** Not applicable.

**Conflicts of Interest:** The authors declare no conflict of interest.

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
