# Peer review of "On the Inner Horizon Instability of Non-Singular Black Holes"

_universe, doi:10.3390/universe8040204_

Round 1

Reviewer 1 Report

In this manuscript authors aim to explore the viability of regular black holes in connection to the inner horizon instability. The model is simplistic in the sense that effects responsible for singularity resolution do not affect the spacetime outside the horizon. Two types of perturbations are studied: (i)  when the spacetime is perturbed by two null shells crossing near the inner horizon (ii)  physical perturbation arising from a continuous flux of energy. Using these exercises, authors show that non-perturbative backreaction can be important and is needed to be studied carefully to construct regular black hole models. Since the results assume no dynamics, I find these results quite interesting and useful.

The paper is well written, however it will benefit by:

  1. Stating the conclusion clearly in the last section in a separate para.
  2. Briefly expanding on the approach in refs 8 and 9 in Sec 2 and Ref 7 near eq. 21

Author Response

We would like to thank the referee for their comments that helped us improve the clarity of the paper. 

According to the suggestions, we have

  1. Separated subsection IV.7 from the rest of section IV. In this subsection we provide a summary of our main conclusions;
  2. We have expanded on the approach in Sec 2 by adding eq. (9) and (11). We believe that these additions improve the clarity of the section. On the other hand, the analysis around Eq. 21 is purely technical and it is explained in detail in our previous work Ref. 8 (Ref. 7 of the previous version of the draft). Therefore, unless the referee feels strongly about this point, we would prefer to avoid adding the technical details in this section.

Reviewer 2 Report

It is recognized that this article is written as a Conference Report. 

It is considered that for regular black holes, the classical singularity is replaced by a non-singular core without necessarily modifying the spacetime outside the trapping horizon. 
In this proceedings manuscript, the physical mechanism leading to the instability of the central core is reviewed. In particular, it is argued that non-perturbative backreation is non-negligible and must be taken into account to provide a meaningful description of physical black holes.

The discussions could be interesting and the mathematical explanations might be useful for the related works. Thus, this proceedings manuscript could be worthy of being published. 

Before publication, e.g., in Introductory part, the original work published in a scientific journal for this conference report must be cited explicitly. Moreover, English wordings should be rechecked. 

Author Response

We thank the referees for their comments. 

At the end of the introduction, we explicitly mention that the work is based mainly on Refs. [7,8].

We had the draft spellchecked by a native English speaker, and we have done minor modifications to the English wordings. 

Reviewer 3 Report

The manuscript studies the inner horizon instability of non-singular black holes, which provides important clarifications on the subject. The paper should be published.

My only minor objection: I think the authors should be more fair regarding the statement that a full theory of quantum gravity will not produce a singularity. This may be true, but there is no proof of this "hope", and there are indeed other points of view [https://arxiv.org/abs/2112.08531 gives some alternatives].

Another question that I have is the following: in the context of analog black holes, for example, those studied by Steinhauer and coauthors [e.g. 2110.06796], there is also an inner horizon. Of course in the analog system there is no singularity, so it is "regular". However it seems that inner horizons in such a system is stable (this I have asked Steinhauer and he confirmed it as such). Can the authors comment on this? Maybe we are pushing analogy too far? But then what sort of physical properties do we expect to carry over from the analog side to the actual gravity side? Since this is more like a side comment, it is sufficient that the authors give a short comment on this issue. 

Author Response

We would like to thank the referee for their interesting comments. 

In response to these comments, we have

  1. Added a comment in the introduction referring the readers to https://arxiv.org/abs/2112.08531 for some alternatives points of view regarding singularities regularization.
  2. Regarding analog black holes, the main point is that our analysis cannot be straightforwardly applied to Lorentz violating systems. In the manuscript, we have added Sec. IV.5 to address this point extensively.